# Nearly Isometric Embedding by Relaxation

**James McQueen**
Department of Statistics
University of Washington
Seattle, WA 98195
jmcq@u.washington.edu

**Marina Meilă**
Department of Statistics
University of Washington
Seattle, WA 98195
mmp@stat.washington.edu

**Dominique Perrault-Joncas**
Google
Seattle, WA 98103
dcpjoncas@gmail.com

## Abstract

Many manifold learning algorithms aim to create embeddings with low or no distortion (isometric). If the data has intrinsic dimension $d$, it is often impossible to obtain an isometric embedding in $d$ dimensions, but possible in $s > d$ dimensions. Yet, most geometry preserving algorithms cannot do the latter. This paper proposes an embedding algorithm to overcome this. The algorithm accepts as input, besides the dimension $d$, an embedding dimension $s \geq d$. For any data embedding **Y**, we compute a Loss(**Y**), based on the push-forward Riemannian metric associated with **Y**, which measures deviation of **Y** from from isometry. Riemannian Relaxation iteratively updates **Y** in order to decrease Loss(**Y**). The experiments confirm the superiority of our algorithm in obtaining low distortion embeddings.

## 1   Introduction, background and problem formulation

Suppose we observe data points sampled from a smooth manifold $\mathcal{M}$ with intrinsic dimension $d$ which is itself a submanifold of $D$-dimensional Euclidean space $\mathcal{M} \subset \mathbb{R}^D$. The task of manifold learning is to provide a mapping $\phi : \mathcal{M} \to \mathcal{N}$ (where $\mathcal{N} \subset \mathbb{R}^s$) of the manifold into lower dimensional space $s \ll D$. According to the Whitney Embedding Theorem [11] we know that $\mathcal{M}$ can be embedded smoothly into $\mathbb{R}^{2d}$ using one homeomorphism $\phi$. Hence we seek one smooth map $\phi : \mathcal{M} \to \mathbb{R}^s$ with $d \leq s \leq 2d \ll D$.

Smooth embeddings preserve the topology of the original $\mathcal{M}$. Nevertheless, in general, they distort the geometry. Theoretically speaking[1], preserving the geometry of an embedding is embodied in the concepts of *Riemannian metric* and *isometric embedding*. A Riemannian metric $g$ is a symmetric positive definite tensor field on $\mathcal{M}$ which defines an inner product $<,>_g$ on the tangent space $\mathcal{T}_p\mathcal{M}$ for every point $p \in \mathcal{M}$. A *Riemannian manifold* is a smooth manifold with a Riemannian metric at every point. A diffeomorphism $\phi : \mathcal{M} \to \mathcal{N}$ is called an isometry iff for all $p \in \mathcal{M}, u, v \in \mathcal{T}_p\mathcal{M}$ we have $< u, v >_{g_p} = < d\phi_p u, d\phi_p v >_{h_{\phi(p)}}$. By Nash's Embedding Theorem [13], it is known that any smooth manifold of class $C^k, k \geq 3$ and intrinsic dimension $d$ can be embedded isometrically in the Euclidean space $\mathbb{R}^s$ with $s$ polynomial in $d$.

In unsupervised learning, it is standard to assume that $(\mathcal{M}, g_0)$ is a submanifold of $\mathbb{R}^D$ and that it inherits the Euclidean metric from it[2]. An embedding $\phi : \mathcal{M} \to \phi(\mathcal{M}) = \mathcal{N}$ defines a metric $g$ on $\mathcal{N}$ given by $< u, v >_{g(\phi(p))} = < d\phi^{-1}u, d\phi^{-1}v >_{g_0(p)}$ called the *pushforward* Riemannian metric; $(\mathcal{M}, g_0)$ and $(\mathcal{N}, g)$ are isometric.

Much previous work in non-linear dimension reduction[16, 20, 19] has been driven by the desire to find smooth embeddings of low dimension that are isometric in the limit of large $n$. This work has met with mixed success. There exists the constructive implementation [19] of Nash's proof

technique, which guarantees consistence and isometry. However, the algorithm presented falls short of being practical, as the embedding dimension $s$ it requires is significantly higher than the minimum necessary, a major drawback in practice. Overall, the algorithm leads to mappings $\phi$ that, albeit having the desired properties, are visually unintuitive, even for intrinsic dimensions as low as $d = 1$.

There are many algorithms, too many for an exhaustive list, which map the data using a cleverly chosen reconstruction criterion. The criterion is chosen so that the mapping $\phi$ can be obtained as the unique solution of a "classic" optimization problem, e.g. Eigendecomposition for Laplacian Eigenmaps [2], Diffusion Maps [12] and LTSA [21], Semidefinite Programming for Maximum Variance Unfolding [20] or Multidimensional Scaling for Isomap [3]. These embedding algorithms sometimes come with guarantees of consistency [2] and, only in restricted cases, isometry [3].

In this paper we propose an approach which departs from both these existing directions. The main difference, from the algorithmic point of view, is that the loss function we propose does not have a form amenable to a standard solver (and is not even guaranteed to be convex or unimodal). Thus, we do not obtain a mapping $\phi$ in "one shot", as the previous algorithms do, but by the gradual improvements of an initial guess, i.e. by gradient descent. Nevertheless, the loss we define directly measures the deviation from isometry; therefore, when this loss is (near) 0, (near) isometry is achieved.

The algorithm is initialized with a smooth embedding $\mathcal{Y} = \phi(\mathcal{M}) \subseteq \mathbb{R}^s$, $s \geq d$; we define the objective function $\text{Loss}(\mathcal{Y})$ as the averaged deviation of the pushforward metric from isometry. Then $\mathcal{Y}$ is iteratively changed in a direction that decreases Loss. To construct this loss function, we exploit the results of [15] who showed how a pushforward metric can be estimated, for finite samples and in any given coordinates, using a discrete estimator of the Laplace-Beltrami operator $\Delta_{\mathcal{M}}$. The optimization algorithm is outlined in Algorithm 1.

---

**Input** : data $\mathbf{X} \in \mathbb{R}^{n \times D}$, kernel function $K_h()$, weights $w_{1:n}$, intrinsic dimension $d$, embedding dimension $s$
        Initial coordinates $\mathbf{Y} \in \mathbb{R}^{n \times s}$, with $\mathbf{Y}_{k,:}$ representing the coordinates of point $k$.
**Init** : Compute Laplacian matrix $\mathcal{L} \in \mathbb{R}^{n \times n}$ using $\mathbf{X}$ and $K_h()$.
**while** *not converged* **do**
       Compute $\mathbf{H} = [\mathbf{H}_k]_{k=1:n} \in \mathbb{R}^{n \times s \times s}$ the (dual) pushforward metric at data points from $\mathbf{Y}$ and $\mathcal{L}$.
       Compute $\text{Loss}(\mathbf{H}_{1:n})$ and $\nabla_{\mathbf{Y}} \text{Loss}(\mathbf{H})$
       Take a gradient step $\mathbf{Y} \leftarrow \mathbf{Y} - \eta \nabla_{\mathbf{Y}} \text{Loss}(\mathbf{H})$
**end**
**Output** : $\mathbf{Y}$

**Algorithm 1:** Outline of the Riemannian Relaxation Algorithm.

---

A remark on notation is necessary. Throughout the paper, we denote by $\mathcal{M}, p \in \mathcal{M}, \mathcal{T}_p \mathcal{M}, \Delta_{\mathcal{M}}$ a manifold, a point on it, the tangent subspace at $p$, and the Laplace-Beltrami operator in the abstract, coordinate free form. When we describe algorithms acting on data, we will use coordinate and finite sample representations. The data is $\mathbf{X} \in \mathbb{R}^{n \times D}$, and an embedding thereof is denoted $\mathbf{Y} \in \mathbb{R}^{n \times s}$; rows $k$ of $\mathbf{X}, \mathbf{Y}$, denoted $\mathbf{X}_k, \mathbf{Y}_k$ are coordinates of data point $k$, while the columns, e.g $\mathbf{Y}^j$ represent functions of the points, i.e restrictions to the data of functions on $\mathcal{M}$. The construction of $\mathcal{L}$ (see below) requires a *kernel*, which can be the (truncated) gaussian kernel $K_h(z) = \exp(z^2/h)$, $|z| < rh$ for some fixed $r > 0$ [9, 17]. Besides these, the algorithm is given a set of *weights* $w_{1:n}$, $\sum_k w_k = 1$.

The construction of the loss is based on two main sets of results that we briefly review here. First, an estimator $\mathcal{L}$ of the *Laplace-Beltrami* operator $\Delta_{\mathcal{M}}$ of $\mathcal{M}$, and second, an estimator of the pushforward metric $g$ in the current coordinates $\mathbf{Y}$.

To construct $\mathcal{L}$ we use the method of [4], which guarantees that, if the data are sampled from a manifold $\mathcal{M}$, $\mathcal{L}$ converges to $\Delta_{\mathcal{M}}$ [9, 17]. Given a set of points in high-dimensional Euclidean space $\mathbb{R}^D$, represented by the $n \times D$ matrix $\mathbf{X}$, construct a weighted *neighborhood graph* $\mathcal{G} = (\{1 : n\}, W)$ over them, with $\mathbf{W} = [W_{kl}]_{k,l=1:n}$. The weight $W_{kl}$ between $\mathbf{X}_{k:}$ and $\mathbf{X}_{l:}$ is the *heat kernel* [2] $W_{kl} \equiv K_h(||\mathbf{X}_{k:} - \mathbf{X}_{l:}||)$ with $h$ a *bandwidth* parameter fixed by the user, and $||\ ||$ the Euclidean norm. Next, construct $\mathcal{L} = [\mathcal{L}_{kl}]_{ij}$ of $\mathcal{G}$ by

$$\mathbf{D} = \text{diag}(\mathbf{W1}), \quad \tilde{\mathbf{W}} = \mathbf{D}^{-1} \mathbf{W} \mathbf{D}^{-1}, \quad \tilde{\mathbf{D}} = \text{diag}(\tilde{\mathbf{W}}\mathbf{1}), \text{ and } \mathcal{L} = \tilde{\mathbf{D}}^{-1} \tilde{\mathbf{W}} \tag{1}$$

Equation (1) represents the discrete versions of the renormalized Laplacian construction from [4]. Note that $\mathbf{W}, \mathbf{D}, \tilde{\mathbf{D}}, \tilde{\mathbf{W}}, \mathcal{L}$ all depend on the bandwidth $h$ via the heat kernel. The consistency of $\mathcal{L}$ has been proved in e.g [9, 17].

The second fact we use is the relationship between the Laplace-Beltrami operator and the Riemannian metric on a manifold [11]. Based on this, [15] gives a a construction method for a discrete estimator of the Riemannian metric $g$, in any given coordinate system, from an estimate $\mathcal{L}$ of $\Delta_{\mathcal{M}}$. In a given coordinate representation $\mathbf{Y}$, a Riemannian metric $g$ at each point is an $s \times s$ positive semidefinite matrix of rank $d$. The method of [15] obtains the matrix Moore-Penrose pseudoinverse of this metric (which must be therefore inverted to obtain the pushforward metric). We denote this inverse at point $k$ by $\mathbf{H}_k$; let $\mathbf{H} = [\mathbf{H}_k, \ k = 1, \ldots n]$ be the three dimensional array containing the inverse for each data point. Note that $\mathbf{H}$ is itself the (discrete estimate of) a Riemannian metric, called the *dual* (pushforward) metric. With these preliminaries, the method of [15] computes $\mathbf{H}$ by

$$\mathbf{H}^{ij} \;=\; \frac{1}{2}\left[\mathcal{L}(\mathbf{Y}^i \cdot \mathbf{Y}^j) - \mathbf{Y}^i \cdot (\mathcal{L}\mathbf{Y}^j) - \mathbf{Y}^j \cdot (\mathcal{L}\mathbf{Y}^i)\right] \tag{2}$$

Where here $\mathbf{H}^{ij}$ is the vector whose $k$th entry is the $ij$th element of the dual pushforward metric $\mathbf{H}$ at the point $k$ and $\cdot$ denotes element-by-element multiplication.

## 2  The objective function Loss

**The case $s = d$ (embedding dimension equals intrinsic dimension).**   Under this condition, it can be shown [10] that $\phi : \mathcal{M} \to \mathbb{R}^d$ is an isometry iff $g_p, \ p \in \mathcal{M}$ expressed in a normal coordinate system equals the unit matrix $\mathbf{I}_d$. Based on this observation, it is natural to measure the quality of the data embedding $\mathbf{Y}$ as the departure of the Riemannian metric obtained via (2) from the unit matrix.

This is the starting idea for the distortion measure we propose to optimize. We develop it further as follows. First, we choose to use the dual of $g$, evaluated by $\mathbf{H}$ instead of pushforward metric itself. Naturally $\mathbf{H}_k = \mathbf{I}_d$ iff $\mathbf{H}_k^{-1} = \mathbf{I}_d$, so the dual metric identifies isometry as well. When no isometric transformation exists, it is likely that optimizing w.r.t $g$ and optimizing w.r.t $h$ will arrive to different embeddings.    There is no mathematically compelling reason, however, to prefer optimizing one over the other. We choose to optimize w.r.t $h$ for three reasons; (1) it is computationally faster, (2) it is numerically more stable, and (3) in our experience users find $\mathbf{H}$ more interpretable. [3]

Second, we choose to measure the distortion of $\mathbf{H}_k$ by $||\mathbf{H}_k - \mathbf{I}||$ where $||\ ||$ denotes the matrix spectral norm. This choice will be motivated shortly. Third, we choose the weights $w_{1:n}$ to be proportional to $\tilde{\mathbf{D}}$ from (1). As [4] show, these values converge to the sampling density $\pi$ on $\mathcal{M}$. Putting these together, we obtain the loss function

$$\text{Loss}(\mathbf{Y}; \mathcal{L}, w) = \sum_{k=1}^{n} w_k \left|\left|\mathbf{H}_k - \mathbf{I}_d\right|\right|^2 . \tag{3}$$

To motivate the choice of a "squared loss" instead of simply using $||\mathbf{H}_k - \mathbf{I}_d||$, notice (the proofs are straightforward) that $||\ ||$ is not differentiable at 0, but $||\ ||^2$ is.

A natural question to ask about Loss is if it is convex. The following proposition proved in the Supplement summarizes a set of relevant convexity facts.

**Proposition 1** *Denote by $\lambda_{1:d}(\mathbf{H}_k) \geq 0$ the eigenvalues of $\mathbf{H}_k$, in decreasing order and assume $\mathbf{Y}$ is in a compact, convex set. Then*
*1. $\lambda_1(\mathbf{H}_k)$, $\lambda_1(\mathbf{H}_k) - \lambda_d(\mathbf{H}_k)$ and $\lambda_1(\mathbf{H}_k) - \sum_{d'=1}^{d} \lambda_{d'}(\mathbf{H}_k)$ are convex in $\mathbf{Y}$.*
*2. $||\mathbf{H}_k - \mathbf{I}_d||$ is convex in $\mathbf{Y}$ for $(\lambda_1(\mathbf{H}_k) + \lambda_d(\mathbf{H}_k))/2 \geq 1$ and concave otherwise.*
*3. $||\mathbf{H}_k - \mathbf{I}_d||^2$ is convex in $\mathbf{Y}$ whenever $||\mathbf{H}_k - \mathbf{I}_d||$ is convex and differentiable in $\mathbf{Y}$.*

This proposition shows that Loss may not be convex near its minimum, and moreover that squaring the loss only improves convexity.

**Choosing the right measure of distortion**   The *norm of a Hermitian bilinear functional (i.e symmetric tensor of order 2)* $g : \mathbb{R}^s \times \mathbb{R}^s \to \mathbb{R}$ is defined as $\sup_{u\neq 0} |g(u,u)|/||u||$. In a fixed orthonormal base of $\mathbb{R}^s$, $g(u,v) = u'\mathbf{G}v$, $||g|| = \sup_{u\neq 0} |u'\mathbf{G}u|$. One can define norms with respect to any metric $g_0$ on $\mathbb{R}^s$ (where $g_0$ is represented in coordinates by $\mathbf{G}_0$, a symmetric, positive definite matrix), by $||u||_{\mathbf{G}_0} = u'\mathbf{G}_0 u$, respectively $||g||_{\mathbf{G}_0} = \sup_{u\neq 0} |u'\mathbf{G}u|/||u||_{\mathbf{G}_0} =$

$\sup_{\tilde{u}\neq 0}|\tilde{u}'\mathbf{G}_0^{-1/2}\mathbf{GG}_0^{-1/2}\tilde{u}|/||\tilde{u}|| = \lambda_{max}(\mathbf{G}_0^{-1/2}\mathbf{GG}_0^{-1/2})$. In particular, since any Riemannian metric at a point $k$ is a $g$ as above, setting $g$ and $g_0$ respectively to $\mathbf{H}_k$ and $\mathbf{I}_d$ we measure the *operator* norm of the distortion by $||\mathbf{H}_k - \mathbf{I}_d||$. In other words, the appropriate operator norm we seek can be expresed as a matrix spectral norm.

The expected loss over the data set, given a distribution represented by the weights $w_{1:n}$ is then identical to the expression of Loss in (3). If the weights are computed as in (1), it is easy to see that the loss function in (3) is the finite sample version of the squared $L_2$ distance between $h$ and $g_0$ on the space of Riemannian metrics on $\mathcal{M}$, w.r.t base measure $\pi dV_{g_0}$

$$||h - g_0||_{g_0}^2 = \int_{\mathcal{M}} ||h - g_0||_{g_0}^2 \pi dV_{g_0}, \quad \text{with } dV_{g_0}\text{volume element on } \mathcal{M}. \tag{4}$$

**Defining** Loss **for embeddings with $s > d$ dimensions**  Consider $\mathbf{G}, \mathbf{G}_0 \in \mathbb{R}^{s\times s}$, two symmetric matrices with $\mathbf{G}_0$ semipositive definite of rank $d < s$. We would like to extend the $\mathbf{G}_0$ norm of $\mathbf{G}$ to this case. We start with the family of norms $||||_{\mathbf{G}_0+\varepsilon\mathbf{I}_s}$ for $\epsilon > 0$ and we define

$$||\mathbf{G}||_{\mathbf{G}_0} = \lim_{\epsilon\to 0} ||\mathbf{G}||_{\mathbf{G}_0+\varepsilon\mathbf{I}_s}. \tag{5}$$

**Proposition 2** *Let $\mathbf{G}, \mathbf{G}_0 \in \mathbb{R}^{s\times s}$ be symmetric matrices, with $\mathbf{G}_0$ semipositive definite of rank $d < s$, and let $\epsilon > 0$, $\gamma(u, \varepsilon) = \frac{u'\mathbf{G}u}{u'\mathbf{G}_0 u+\epsilon ||u||^2}$. Then,*

1. *$||\mathbf{G}||_{\mathbf{G}_0+\varepsilon\mathbf{I}_s} = ||\tilde{\mathbf{G}}||_2$ with $\tilde{\mathbf{G}} = (\mathbf{G}_0 + \epsilon I)^{-1/2}\mathbf{G}(\mathbf{G}_0 + \epsilon I)^{-1/2}$.*

2. *If $||\mathbf{G}||_{\mathbf{G}_0+\varepsilon\mathbf{I}_s} < r$, then $\lambda^{\dagger}(\mathbf{G}) < \epsilon r$ with $\lambda^{\dagger}(\mathbf{G}) = \sup_{v\in\text{Null}(\mathbf{G}_0)}\gamma(v, \varepsilon)$,*

3. *$||||_{\mathbf{G}_0}$ is a matrix norm that takes infinite values when $\text{Null } \mathbf{G}_0 \not\subseteq \text{Null } \mathbf{G}$.*

Hence, $||||_{\mathbf{G}_0+\varepsilon\mathbf{I}_s}$ can be computed as the spectral norm of a matrix. The computation of $||||_{\mathbf{G}_0}$ is similar, with the additional step of checking first if $\text{Null } \mathbf{G}_0 \not\subseteq \text{Null } \mathbf{G}$, in which case we output the value $\infty$. Let $B_\epsilon(\mathbf{0}, r)$ $(B(\mathbf{0}, r))$ denote the $r$-radius ball centered at $\mathbf{0}$ in the $||||_{\mathbf{G}_0+\varepsilon\mathbf{I}_s}$ $(||||_{\mathbf{G}_0})$. From Proposition 2 it follows that if $\mathbf{G} \in B_\epsilon(\mathbf{0}, r)$ then $\lambda^{\dagger}(\mathbf{G}) < \epsilon r$ and if $\mathbf{G} \in B(\mathbf{0}, r)$ then $\text{Null}(\mathbf{G}_0) \subseteq \text{Null}(\mathbf{G})$. In particular, if $\text{rank } \mathbf{G} = \text{rank } \mathbf{G}_0$ then $\text{Null}(\mathbf{G}) = \text{Null}(\mathbf{G}_0)$.

To define the loss for $s > d$ we set $\mathbf{G} = \mathbf{H}_k$ and $\mathbf{G}_0 = \mathbf{U}_k\mathbf{U}_k'$, with $\mathbf{U}_k$ an orthonormal basis for $\mathcal{T}_k\mathcal{M}$ the tangent subspace at $k$. The norms $||||_{\mathbf{G}_0+\varepsilon\mathbf{I}_s}, ||||_{\mathbf{G}_0}$ act as soft and hard barrier functions constraining the span of $\mathbf{H}_k$ to align with the tangent subspace of the data manifold.

$$\text{Loss}(\mathbf{Y}; \mathcal{L}, w, d, \varepsilon_{orth}) = \sum_{k=1}^{n} w_k|| \underbrace{(\mathbf{U}_k\mathbf{U}_k' + \varepsilon_{orth}^2\mathbf{I}_s)^{-1/2}\left(\mathbf{H}_k - \mathbf{U}_k\mathbf{U}_k'\right)(\mathbf{U}_k\mathbf{U}_k' + \varepsilon_{orth}^2\mathbf{I}_s)^{-1/2}}_{\tilde{\mathbf{G}}_k} ||^2.$$

$$\tag{6}$$

## 3   Optimizing the objective

Let $\mathcal{L}_k$ denote the $k$th row of $\mathcal{L}$, then $\mathbf{H}_k$ can be rewritten in the convenient form

$$\mathbf{H}_k(\mathbf{Y}) = \frac{1}{2}\mathbf{Y}'[\text{trace}(\mathcal{L}_k) - (e_k e_k'\mathcal{L}) - (e_k e_k'\mathcal{L})']\mathbf{Y} \equiv \frac{1}{2}\mathbf{Y}'\mathbf{L}_k\mathbf{Y} \tag{7}$$

where $e_k$ refers to the $k$th standard basis vector of $\mathbb{R}^n$ and $\mathbf{L}_k$ is a symmetric positive semi-definite matrix precomputed from entries in $\mathcal{L}$; $\mathbf{L}_k$ has non-zero rows only for the neighbors of $k$.

**Proposition 3** *Let $\text{Loss}_k$ denote term $k$ of Loss. If $s = d$, the gradient of $\text{Loss}_k$ as given by (3) is*

$$\frac{\partial \text{Loss}_k}{\partial \mathbf{Y}} = 2w_k\lambda_k^*\mathbf{L}_k\mathbf{Y}\mathbf{u}_k\mathbf{u}_k', \tag{8}$$

*with $\lambda_k^*$ the largest eigenvalue of $\mathbf{H}_k - \mathbf{I}_d$ and $\mathbf{u}_k$ is the corresponding eigenvector.*
*If $s > d$, the gradient of $\text{Loss}_k$ of (6) is*

$$\frac{\partial \text{Loss}_k}{\partial \mathbf{Y}} = 2w_k\lambda_k^*\mathbf{L}_k\mathbf{Y}\mathbf{\Pi}_k\mathbf{u}_k\mathbf{u}_k'\mathbf{\Pi}_k' \tag{9}$$

*where $\mathbf{\Pi}_k = (\mathbf{U}_k\mathbf{U}_k' + (\varepsilon_{orth})_k\mathbf{I}_s)^{-1/2}$, $\lambda_k^*$ is the largest eigenvalue of $\tilde{\mathbf{G}}_k$ of (6) and $\mathbf{u}_k$ is the corresponding eigenvector.*

When embedding in $s > d$ dimensions, the loss function depends at each point $k$ on finding the $d$-dimensional subspace $\mathbf{U}_k$. Mathematically, this subspace coincides with the span of the Jacobian $D\mathbf{Y}_k$ which can be identified with the $d$-principal subspace of $\mathbf{H}_k$. When computing the gradient of Loss we assume that $\mathbf{U}_{1:n}$ are fixed. Since the derivatives w.r.t $\mathbf{Y}$ are taken only of $\mathbf{H}$ and not of the tangent subspace $\mathbf{U}_k$, the algorithm below is actually an alternate minimization algorithm, which reduces the cost w.r.t $\mathbf{Y}$ in one step, and w.r.t $\mathbf{U}_{1:n}$ in the alternate step.

## 3.1 Algorithm

We optimize the loss (3) or (6) by projected gradient descent with line search (subject to the observation above). The projection consists of imposing $\sum_k \mathbf{Y}_k = 0$, which we enforce by centering $\nabla \mathbf{Y}$ before taking a step. This eliminates the degeneracy of the Loss in (3) and (6) w.r.t constant shift in $\mathbf{Y}$. To further improve the good trade-off between time per iteration and number of iterations, we found that a heavy-ball method with parameter $\alpha$ is effective. At each iteration computing the gradient is $\mathcal{O}((S + s^3)n)$ where $S$ is the number of nonzero entries of $\mathcal{L}$.

---

**Input** : data $\mathbf{X}$, kernel function $K_h()$, initial coordinates $\mathbf{Y}^0$, weights $w_{1:n}$, intrinsic dimension $d$,
        orthonormal tolerance $\varepsilon_{orth}$, heavy ball parameter $\alpha \in [0, 1)$
**Init**    : Compute: graph Laplacian $\mathcal{L}$ by (1), matrices $\mathbf{L}_{1:n}$ as in (7). Set $\mathbf{S} = 0$
**while** *not converged* **do**
     Compute $\nabla$ Loss:
     **for** *all $k$* **do**
               1. Calculate $\mathbf{H}_k$ via (2);
               2. **If** $s > d$
                   (a) Compute $\mathbf{U}_k$ by SVD from $\mathbf{H}_k$;
                   (b) Compute gradient of $\nabla \text{Loss}_k(\mathbf{Y})$ using (9);
               3. **Else** ($s = d$): calculate gradient $\nabla \text{Loss}_k(\mathbf{Y})$ using (8);
               4. Add $\nabla \text{Loss}_k(\mathbf{Y})$ to the total gradient;
     **end**
     Take a step in $\mathbf{Y}$:
               1. Compute projected direction $\mathbf{S}$ and project $\mathbf{S} \leftarrow (\mathbf{I}_n - e_n e_n') \nabla \text{Loss} + \alpha \mathbf{S}$;
               2. Find step size $\eta$ by line search and update $\mathbf{Y} \leftarrow \mathbf{Y} - \eta \mathbf{S}$;
**end**
**Output** : $\mathbf{Y}$

**Algorithm 2:** RIEMANNIANRELAXATION (RR)

---

## 3.2 For large or noisy data

Here we describe an extension of the RR Algorithm which can naturally adapt to large or noisy data, where the manifold assumption holds only approximately. The idea is to subsample the data, but in a highly non-uniform way that improves the estimation of the geometry.

A simple peliminary observation is that, when an embedding is smooth, optimizing the loss on a subset of the data will be sufficient. Let $\mathcal{I} \subset \{1, \dots n\}$ be set of size $n' < n$. The subsampled loss $\text{Loss}_{\mathcal{I}}$ will be computed only for the points $k' \in \mathcal{I}$. If every point $k$ has $\mathcal{O}(d)$ neighbors in $\mathcal{I}$, this assures that the gradient of $\text{Loss}_{\mathcal{I}}$ will be a good approximation of $\nabla$ Loss at point $k$, even if $k \notin \mathcal{I}$, and does not have a term containing $\mathbf{H}_k$ in $\text{Loss}_{\mathcal{I}}$. To optimize $\text{Loss}_{\mathcal{I}}$ by RR, it is sufficient to run the "for" loop over $k' \in \mathcal{I}$. Algorithm PCS-RR below describes how we choose a "good" subsample $\mathcal{I}$, with the help of the PRINCIPALCURVES algorithm of [14].

---

**Input**   : data $\mathbf{X}$, kernel function $K_h()$, initial coordinates $\mathbf{Y}^0$, intrinsic dimension $d$, subsample size $n'$, other
         parameters for RR
Compute $\hat{\mathbf{X}} = \text{PRINCIPALCURVES}(\mathbf{X}, K_h, d)$
Take a uniform sample $\mathcal{I}_0$ of size $n'$ from $\{1, \dots n\}$ (without replacement).
**for** *$k'$ in $\mathcal{I}_0$* **do**
    Find $\mathbf{X}_l$ the nearest neigbor in $\mathbf{X}$ of $\hat{\mathbf{X}}_{k'}$, and add $l$ to $\mathcal{I}$ (removing duplicates)
**end**
**Output** : $\mathbf{Y} = \text{RR}(\mathbf{Y}^0, K_h, d, \mathcal{I}, \dots)$

**Algorithm 3:** PRINCIPALCURVES-RIEMANNIANRELAXATION (PCS-RR)

---

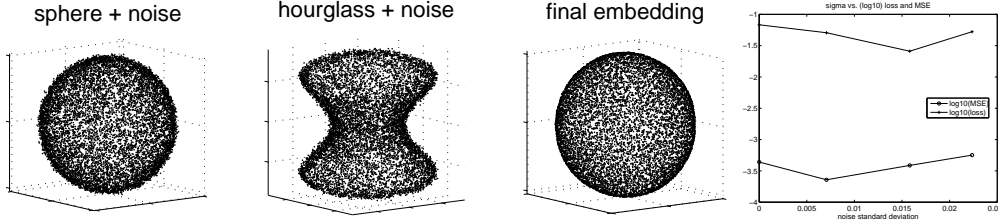

Figure 1: **Hourglass to sphere.** From left to right: target **Y** (noisy sphere), initialization $\mathbf{Y}^0$ of RR (noisy hourglass), output of RR, mean-squared error and Loss vs. noise level $\sigma$ (on a $\log_{10}$ scale). Convergence of RR was achieved after 400 iterations.

Informally speaking, PRINCIPALCURVES uses a form of Mean-Shift to obtain points in the $d$-dimensional manifold of highest density in the data. The result is generally biased, however [7] have shown that this algorithm offers a very advantageous bias-variance trade-off in case of manifolds with noise. We use the output $\hat{\mathbf{Y}}$ of PRINCIPALCURVES to find a subset of points that (1) lie in a high density region relative to most directions in $\mathbb{R}^D$ and (2) are "in the middle" of their neighbors, or more formally, have neighborhoods of dimension at least $d$. In other words, this is a good heuristic to avoid "border effects", or other regions where the $d$-manifold assumption is violated.

## 4 Experimental evaluation

**Hourglass to sphere** illustrates how the algorithm works for $s = 3$, $d = 2$. The data **X** is sampled uniformly from a sphere of radius 1 with intrinsic dimension $d = 2$. We sample $n = 10000$ points from the sphere and add i.i.d. Gaussian noise with $\Sigma = \sigma^2/s\mathbf{I}_s$[4], estimating the Laplacian $\mathcal{L}$ on the noisy data **X**. We initialize with a noisy "hourglass" shape in $s = 3$ dimensions, with the same noise distribution as the sphere. If the algorithm works correctly, by using solely the Laplacian and weights from **X**, it should morph the hourglass $\mathbf{Y}^0$ back into a sphere. The results after convergence at 400 iterations are shown in Fig. 1 (and an animation of this convergence in the Supplement). We see that RR not only recovers the sphere, but it also suppresses the noise.

The next two experiments compare RR to several embedding algorithms w.r.t geometric recovery. The algorithms are Isomap, Laplacian Eigenmaps, HLLE[6], MVU [5] . The embeddings $\mathbf{Y}^{LE,MVU,HLLE}$ need to be rescaled before being evaluated, and we use a Procrustes transformation to the original data. The algorithms are compared w.r.t the dual metric distortion Loss, and w.r.t mean squared errror in pairwise distance (the loss optimized by Isomap [6] ). This is

$$\text{dis}(\mathbf{Y}, \mathbf{Y}^{true}) = 2/n(n-1) \sum_{k \neq k'} \left( ||\mathbf{Y}_k - \mathbf{Y}_{k'}|| - ||\mathbf{Y}_k^{true} - \mathbf{Y}_{k'}^{true}|| \right)^2 \qquad (10)$$

where **Y** is the embedding resulting from the chosen method and $\mathbf{Y}^{true}$ are the true noiseless coordinates. Note that none of Isomap, MVU, HLLE could have been tested on the hourglass to sphere data of the previous example, because they work only for $s = d$. The sample size is $n = 3000$ in both experiments, and noise is added as described above.

**Flat "swiss roll" manifold,** $s = d = 2$. The results are displayed in Fig. 2.

**Curved "half sphere" manifold,** $s = d = 2$. Isometric embedding into 2D is not possible. We examine which of the algorithms achieves the smallest distortions in this scenario. The true distances were computed as arc-lengths on the half-sphere. The results are displayed in Fig 2.

RR was initialized at each method. In almost every initalization and noise level, RR achieves a decrease in dis, in some cases significant decreases. Isomap also performs well and even though RR optimizes a different loss function it never increases dis and often improves on it. This demonstrates the ability of the Riemannian Metric to encode simultaneously all aspects of manifold geom-

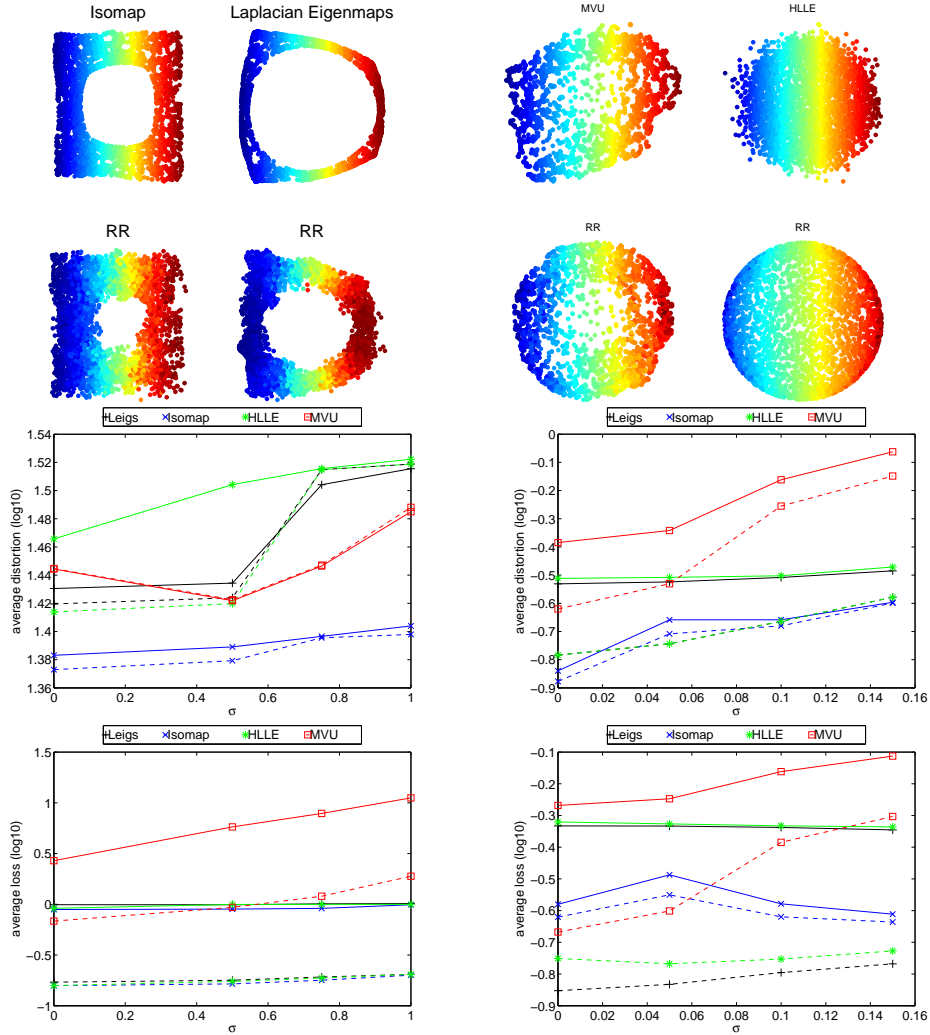

Figure 2: Swiss hole (left) & half sphere (right). Top plots display example initial embeddings and their Riemannian Relaxed versions. Middle row displays dis value vs. noise level $\sigma$. Bottom row displays Loss value vs. noise level $\sigma$. As RR was initialized at each method dashed lines indicated relaxed embeddings

etry. Convergence of RR varies with the initialization but was in all cases faster than Isomap. The extension of RR to PCS-RR allows for scaling to much larger data sets.

## 4.1 Visualizing the main SDSS galaxy sample in spectra space

The data consists of spectra of galaxies from the Sloan Digital Sky Survey[7] [1]. We extracted a subset of spectra whose SNR was sufficiently high, known as the *main sample*. This set contains 675,000 galaxies observed in $D = 3750$ spectral bins, preprocessed by first moving them to a common rest-frame wavelength and filling-in missing data following [18] but using the more sophisticated weighted PCA algorithm of [5], before computing a sparse neighborhood graph and pairwise distances between neighbors in this graph. A log-log plot of the average number neighbors $m(r)$ vs. neighborhood radius $r$ (shown in the Supplement), indicates that the intrinsic dimension of these data varies with the scale $r$. In particular, in order to support $m = O(d)$ neighbors, the radius must be above 60, in which case $d \leq 3$. We embedded the whole data set by Laplacian Eigenmaps, obtaining the graph in Fig. 3 a. This figure strongly suggests that $d$ is not constant for this data cloud, and that the embedding is not isometric (Fig 3, b). We "rescaled" the data along the three evident

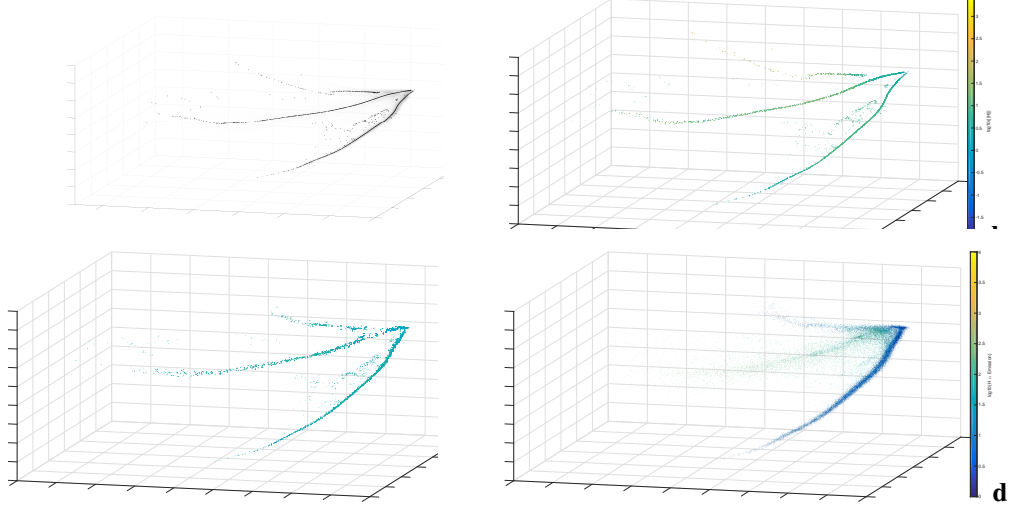

**d**

Figure 3: **a:** Initial LE embedding from $D = 3750$ to $s = 3$ dimensions, with the principal curves $\hat{\mathbf{Y}}$ superimposed. For clarity, we only show a small subsample of the $\mathbf{Y}^0$; a larger one is in the Supplement; **b:** same embedding, only points "on" principal curves, colored by $\log_{10} ||\mathbf{H}_k||$ (hence, 0 represents isometry); **c:** same points as in (b), after RR(color on the same scale as in (b)); **d:** 40,000 galaxies in the coordinates from (c), colored by the strength of Hydrogen $\alpha$ emission, a very nonlinear feature which requires dozens of dimensions to be captured in a linear embedding. Convergence of PCS-RR was achieved after 1000 iterations and took 2.5 hours optimizing a Loss with $n' = 2000$ terms over the $n \times s = 10^5 \times 3$ coordinates, corresponding to the highest density points. (Please zoom for better viewing)

principal curves shown in Figure 3 a by running PCS-RR ($\mathbf{Y}, n = 10^5, n' = 2000, s = 3, d = 1$). In the new coordinates (Fig 3, c), $\mathbf{Y}$ is now close to isometric along the selected curves, while in Fig. 3,b, $||\mathbf{H}_k||$ was in the thousands on the uppermost "arm". This means that, at the largest scale, the units of distance in the space of galaxy spectra are being preserved (almost) uniformly along the sequences, and that they correspond to the distances in the original $D = 3750$ data. Moreover, we expect the distances along the final embedding to be closer on average to the true distance, because of the denoising effect of the embedding. Interpreting the coordinates along these "arms" is in progress. As a next step of the analysis, RR with $s = d = 3$ will be used to rescale the high-density region at the confluence of the three principal curves.

## 5 Discussion

Contributions: we propose a new, natural, way to measure the distortion from isometry of any embedding $\mathbf{Y} \in \mathbb{R}^{n \times s}$ of a data set $\mathbf{X} \in \mathbb{R}^{n \times D}$, and study its properties. The distortion loss is based on an estimate of the push-forward Riemannian metric into Euclidean space $\mathbb{R}^s$.

The RR we propose departs from existing non-linear embedding algorithms in several ways. First, instead of a heuristically chosen loss, like pairwise distances, or local linear reconstruction error, it directly optimizes the (dual) Riemannian metric of the embedding $\mathbf{Y}$. When this is successful, and the loss is 0 *all* geometric properties (lengths, angles, volumes) are preserved simultaneously. From the computational point of view, the non-convex loss is optimized iteratively by projected gradient.

Third, our algorithm explicitly requires both an embedding dimension $s$ and an intrinsic dimension $d$ as inputs. Estimating the intrinsic dimension of a data set is not a solved problem, and beyond the scope of this work. However, as a rule of thumb, we propose chosing the smallest $d$ for which Loss is not too large, for $s$ fixed, or, if $d$ is known (something that all existing algorithms assume), increasing $s$ until the loss becomes almost 0. Most existing embedding algorithms, as Isomap, LLE, HLLE, MVU, LTSA only work in the case $s = d$, while Laplacian Eigenmaps/Diffusion Maps requires only $s$ but does not attempt to preserve geometric relations. Finally, RR is computationally competitive with existing algorithms, and can be seamlessly adapted to a variety of situations arising in the analysis of real data sets.

## Footnotes

[1]For a more complete presentation the reader is referred to [8] or [15] or [10].

[2]Sometimes the Riemannian metric on $\mathcal{M}$ is not inherited, but user-defined via a kernel or distance function.

[3]$\mathbf{H}_k$ represents the direction & degree of distortion as opposed to the scaling required to "correct" the space.

[4]For this artificial noise, adding dimensions beyond $s$ has no effect except to increase $\sigma$.

[5]embeddings were computed using drtoolbox: https://lvdmaaten.github.io/drtoolbox/

[6]Isomap estimates the true distances using graph shortest path

[7] www.sdss.org

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
