[Supplementary Material · isometry-nips16-supplement.pdf]

# Near-Isometry by Relaxation: Supplement

## 1 Proof of Proposition 1,3.

We first prove the following Lemma.

**Proposition 1.** *If $f : S \subseteq \mathbb{R}^D \to \mathbb{R}$ is convex, non-negative and $\nabla^2 f$ exists for all $x \in \text{int } S$, then $\frac{1}{2} f^2(x)$ is convex.*

**Proof** $\nabla \left( \frac{1}{2} f^2 \right) = f \nabla f$; $\nabla^2 \left( \frac{1}{2} f^2 \right) = f \nabla^2 f + \nabla f \nabla f'$ which is positive definite whenever $f \nabla^2 f$ is. $\qquad \square$.

Using the above Lemma, and the fact that $||\mathbf{H}_k - \mathbf{I}_n||$ is non-negative and infinitely differentiable almost everywhere , we obtain the desired result. $\qquad \square$

## 2 Proof of Proposition 2

$$
||\mathbf{G}||_{\mathbf{G}_0 + \varepsilon \mathbf{I}_s} \quad = \quad \sup_{u \neq 0} \frac{u' \mathbf{G} u}{u' \mathbf{G}_0 u + \epsilon ||u||^2} \tag{1}
$$

$$
= \quad \sup_{u \neq 0} \frac{v' \mathbf{G}_\epsilon^{-1} \mathbf{G} \mathbf{G}_\epsilon^{-1} v}{||v||^2} \quad \text{with } \mathbf{G}_\epsilon = (\mathbf{G}_0 + \epsilon I)^{1/2} \text{ and } v = \mathbf{G}_\epsilon u \tag{2}
$$

$$
= \quad ||\tilde{\mathbf{G}}||_2 \quad \text{with } \tilde{\mathbf{G}} = (\mathbf{G}_0 + \epsilon I)^{-1/2} \mathbf{G} (\mathbf{G}_0 + \epsilon I)^{-1/2} \tag{3}
$$

For (2), we first prove the following fact

$$
\sup_{u \in \mathbb{R}^s} \frac{|u^T \mathbf{G} u|}{u^T \mathbf{G}_0 u + \epsilon ||u||^2} \begin{cases} = \sup_{u \in \text{Null } \mathbf{G}^\perp} \frac{|u^T \mathbf{G} u|}{u^T \mathbf{G}_0 u + \epsilon ||u||^2} & \text{if Null}(\mathbf{G}) = \text{Null}(\mathbf{G}_0) \\ \leq \max_{\alpha^2 + \beta^2 = 1} \frac{\beta^2 \lambda^\dagger(\mathbf{G}) + \alpha^2 \lambda_{max}(\mathbf{G}) + 2\alpha\beta\Theta_{max}(\mathbf{G},\mathbf{G}_0)}{\beta^2 \epsilon + \alpha^2 (\lambda^*_{min}(\mathbf{G}_0) + \epsilon)} & \text{if Null}(\mathbf{G}) \neq \text{Null}(\mathbf{G}_0) \end{cases} \tag{4}
$$

where $\lambda_{max}(\mathbf{G})$ is the spectral radius of $\mathbf{G}$, $\Theta(\mathbf{G}, \mathbf{G}_0) = \sup_{||u||=||v||=1, v \in \text{Null } \mathbf{G}, u \in \text{Null } \mathbf{G}_0} u' \mathbf{G} v$ is the cosine of the principal angle between $\text{Null } \mathbf{G}$ and $\text{Null } \mathbf{G}_0$, and $\lambda^*_{min}(\mathbf{G}_0)$ is the smallest non-zero eigenvalue of $\mathbf{G}_0$.

Denote for simplicity $g(u) = \frac{|u^T \mathbf{G} u|}{u^T \mathbf{G}_0 u + \epsilon ||u||^2}$. (1) If $\text{Null}(\mathbf{G}) = \text{Null}(\mathbf{G}_0)$ then for $u \in \text{Null } \mathbf{G}$ the value is 0, which cannot be the sup. Let $u_1 = v \oplus u_0$ with $u_0 \in \text{Null } \mathbf{G}$, $v \in \text{Null } \mathbf{G}^\perp$. Then $u_1^T \mathbf{G}_0 u_1 + \epsilon ||u_1||^2 = v^T \mathbf{G}_0 v + \epsilon ||v||^2 + \epsilon ||u_0||^2 > v^T \mathbf{G}_0 v + \epsilon ||v||^2$. Hence, the $u$ which attains the supremum must be in $\text{Null } \mathbf{G}$.

Now note that, if $\text{Null } \mathbf{G} \neq \text{Null } \mathbf{G}_0$, $\mathbb{R}^s = \text{Null } \mathbf{G}_0 \oplus \text{Null } \mathbf{G}_0^\perp$, and $\text{Null } \mathbf{G}_0 = (\text{Null } \mathbf{G}_0 \cap \text{Null } \mathbf{G}) \oplus \mathcal{V}$, with $\mathcal{V}$ the orthogonal complement of $\text{Null } \mathbf{G}_0 \cap \text{Null } \mathbf{G}$ in $\text{Null } \mathbf{G}_0$ and the supremum of $g(u) =$ is attained on $\mathcal{U} = \mathcal{V} \oplus \text{Null } \mathbf{G}_0^\perp$ (as adding any component along the orthogonal complement of this space only adds a positive value to the denominator, increasing $g(u)$). Any $u \in \mathcal{U}$ can be written as $u = \alpha u_0 \oplus \beta v_0$ with $u_0 \in \text{Null } \mathbf{G}_0^\perp$ and $v_0 \in \mathcal{V}$ unit vectors. By upper bounding every term in

the numerator and lower bounding $u_0'\mathbf{G}_0 u_0$ we obtain the result. Note that for $\epsilon$ small enough, the expression in 4 is close to $\frac{1}{\epsilon}\lambda^\dagger(\mathbf{G})$.

For (2), let $v \in \mathcal{V}$ and compute $g(v)$ as above, with $\alpha = 0$. It follows that $g(v) = \frac{|v'\mathbf{G}v|}{\epsilon||v||^2}$ and by taking the supremum over $v \in \mathcal{V}$ we obtain that $\sup_\mathcal{V} g(v) = \frac{1}{\epsilon}\lambda^\dagger(\mathbf{G}) < r$, from which the result follows.

For (3), it is obvious that when $\epsilon \to 0$, $g(v) \to \infty$ on $\mathcal{V}$, but remains finite for $u \notin \mathcal{V}$. More precisely, $||\mathbf{G}||_{\mathbf{G}_0} = \infty$ iff $\mathrm{Null}\,\mathbf{G}_0 \not\subseteq \mathbf{G}$. To verify that $||\,||_{\mathbf{G}_0}$ is a norm, we must verify the triangle inequality, since the other two properties obviously hold. If $||\mathbf{A}||_{\mathbf{G}_0} = \infty$ or $||\mathbf{B}||_{\mathbf{G}_0} = \infty$, triangle inequality holds trivially. Assume then that $||\mathbf{A}||_{\mathbf{G}_0}, ||\mathbf{B}||_{\mathbf{G}_0} < \infty$. Since $||\mathbf{A}||_{\mathbf{G}_0+\varepsilon\mathbf{I}_s} + ||\mathbf{B}||_{\mathbf{G}_0+\varepsilon\mathbf{I}_s} \geq ||\mathbf{A}+\mathbf{B}||_{\mathbf{G}_0+\varepsilon\mathbf{I}_s}$ for every $\epsilon > 0$, then in the limit we will have that $||\mathbf{A}||_{\mathbf{G}_0} + ||\mathbf{B}||_{\mathbf{G}_0} \geq ||\mathbf{A}+\mathbf{B}||_{\mathbf{G}_0}$.

**The norm for comparing Riemannian metric** The *norm of a bilinear functional* $f : \mathbb{R}^2 \times \mathbb{R}^2 \to \mathbb{R}$ is defined as $\sup_{||u||=||v||=1} |f(u,v)|$, or since for a fixed orthonormal base of $\mathbb{R}^s$ $f(u,v) = u'\mathbf{A}v$, $||f|| = \sup_{||u||=||v||=1} |u'\mathbf{A}v|$. If $\mathbf{A}$ is hermitian, then $||f|| = max_{\lambda(\mathbf{A})}|\lambda_i|$ where $\lambda(\mathbf{A})$ denotes the spectrum of $\mathbf{A}$. One can define the norm with respect to any metric $\mathbf{G}_0$ on $\mathbb{R}^s$ where $\mathbf{G}_0$ is a symmetric, positive definite matrix by $||f||_{\mathbf{G}_0} = \sup_{||u||_{\mathbf{G}_0}=||v||_{\mathbf{G}_0}=1} |u'\mathbf{A}v| = \sup_{||\tilde{u}||=||\tilde{v}||=1} |\tilde{u}'\mathbf{G}_0^{-1/2}\mathbf{A}\mathbf{G}_0^{-1/2}\tilde{v}| = max_{\lambda(\mathbf{G}_0^{-1/2}\mathbf{A}\mathbf{G}_0^{-1/2})}|\lambda_i|$ In other words, the appropriate operator norm we seek can be expressed as a (generalized) matrix spectral norm. In our cases $\mathbf{G}_0 = \mathbf{I_d}$ and $\mathbf{A} = \mathbf{H}_k - \mathbf{I}_d$

## 3   Proof of Propositions 3

Note that we can write the loss as:

$$\sum_{k=1}^n \left|\left|\frac{1}{2}\mathbf{\Pi}_k'\mathbf{Y}'\mathbf{L}_k\mathbf{Y}\mathbf{\Pi}_k - \mathbf{\Pi}_k\mathbf{U}_k\mathbf{U}_k\mathbf{\Pi}_k\right|\right|_2^2$$

Where $\mathbf{\Pi}_k = (\mathbf{U}_k\mathbf{U}_k' + (\varepsilon_{orth})_k\mathbf{I}_s)^{-1/2}$. We take the $\mathbf{\Pi}_k$ matrices to be fixed and don't depend on the data points $\mathbf{Y}$ (in practice they do, however, after taking a gradient step we update the $\mathbf{\Pi}_k$ in an E-M style algorithm). Since $\mathbf{U}_k\mathbf{U}_k'$ and $\mathbf{\Pi}_k$ are the identity matrix (the latter multiplied by $1/(1+\varepsilon_{orth})$) when $s = d$ we can compute the derivative when $s > d$ without loss of generality.

### 3.1   Proof of Derivative

Since the derivative is a linear operator it's sufficient to show that the derivative of a single loss function is of the form:

$$\frac{\partial l_k}{\partial Y} = (2|\lambda_k^*|)\mathrm{sgn}(\lambda_k^*)\mathbf{L}_k\mathbf{Y}\mathbf{\Pi}_k\mathbf{u}_k\mathbf{u}_k'\mathbf{\Pi}_k'$$

To compute the derivative we will make use of the chain rule. First define the function $l_k$ as a composition of functions:

$$l_k(\mathbf{Y}) \equiv \rho(P_k(H_k(\mathbf{Y})) - \mathbf{C}_k)$$

With $\mathbf{C}_k = \mathbf{\Pi}_k\mathbf{U}_k\mathbf{U}_k\mathbf{\Pi}_k$ and

$$\rho(\mathbf{U}) = (\max_k |\lambda_k(\mathbf{U})|)^2$$

$$P_k(\mathbf{H}) = \mathbf{\Pi}_k'\mathbf{H}\mathbf{\Pi}_k$$

$$H_k(\mathbf{Y}) = \frac{1}{2}\mathbf{Y}'\mathbf{L}_k\mathbf{Y}$$

Where $\mathbf{U}, \mathbf{H}$ are both symmetric. Here we note that the matrix spectral norm reduces to the spectral radius if $\mathbf{U}$ is symmetric. Since $H_k(\mathbf{Y})$ is defined to be symmetric and $\mathbf{C}_k$ is symmetric this is the case. By the chain rule:

$$Dl_k(\mathbf{Y}) = D\rho(P_k(H_k(\mathbf{Y})) - \mathbf{C}_k)DP_k(H_k(\mathbf{Y}))DH_k(\mathbf{Y})$$

Taking these from left to right:

### 3.1.1 $D\rho$

Since $\rho$ is defined to be the largest (in absolute value) eigenvalue of $\mathbf{U}$ (squared) the derivative[1] is the kronecker product between the corresponding eigenvector and itself multiplied by the sign of the eigenvalue:

$$D\sqrt{\rho(\mathbf{U})} = sgn(\lambda_k^*)(\mathbf{u}_k' \otimes \mathbf{u}_k')$$

Where $|\lambda_k^*| = \sqrt{\rho(\mathbf{U})}$ and $\mathbf{U}\mathbf{u}_k = \lambda_k^* \mathbf{u}_k$ Then since we square the spectral radius we add the factor of $(2|\lambda_k^*|)$ so that:

$$D(\rho(\mathbf{U}) = (2|\lambda_k^*|)sgn(\lambda_k^*)(\mathbf{u}_k' \otimes \mathbf{u}_k')$$

### 3.1.2 $DP_k$

$$DP_k(\mathbf{H}) = (\mathbf{\Pi}_k' \otimes \mathbf{\Pi}_k')$$

*Proof.*

$$P_k(\mathbf{H}) = \mathbf{\Pi}_k' \mathbf{H} \mathbf{\Pi}_k$$
$$dP_k(\mathbf{H}) = \mathbf{\Pi}_k' d\mathbf{H} \mathbf{\Pi}_k$$
$$\Rightarrow vec(dP_k(\mathbf{H})) = vec(\mathbf{\Pi}_k' d\mathbf{H} \mathbf{\Pi}_k)$$
$$= (\mathbf{\Pi}_k' \otimes \mathbf{\Pi}_k')dvec(\mathbf{H})$$

$\square$

### 3.1.3 $DH_k$

$$DH_k(\mathbf{Y}) = \mathbf{N}_s(\mathbf{I}_s \otimes \mathbf{Y}'\mathbf{L}_k)$$

Where $\mathbf{N}_s = \mathbf{I}_{s^2} + \mathbf{K}_{ss}$ for $\mathbf{K}_{ss}$ the commutation matrix defined in Magnus & Neudecker ch. 3 §7.

*Proof.*

$$H_k(\mathbf{Y}) = \frac{1}{2}\mathbf{Y}'\mathbf{L}_k\mathbf{Y}$$

$$\Rightarrow dH_k(\mathbf{Y}) = \frac{1}{2}[(d\mathbf{Y})'\mathbf{L}_k\mathbf{Y} + \mathbf{Y}'\mathbf{L}_k d\mathbf{Y}]$$

$$\Rightarrow vec(dH_k(\mathbf{Y})) = \frac{1}{2}[(\mathbf{Y}'\mathbf{L}_k' \otimes \mathbf{I}_s)dvec(\mathbf{Y}') + (\mathbf{I}_s \otimes \mathbf{Y}'\mathbf{L}_k)dvec(\mathbf{Y})]$$

$$= \frac{1}{2}[(\mathbf{Y}'\mathbf{L}_k' \otimes \mathbf{I}_s)\mathbf{K}_{ns}dvec(\mathbf{Y}) + (\mathbf{I}_s \otimes \mathbf{Y}'\mathbf{L}_k)dvec(\mathbf{Y})]$$

$$= \frac{1}{2}[\mathbf{K}_{ss}(\mathbf{I}_s \otimes \mathbf{Y}'\mathbf{L}_k')dvec(\mathbf{Y}) + (\mathbf{I}_s \otimes \mathbf{Y}'\mathbf{L}_k)dvec(\mathbf{Y})]$$

$$= \frac{1}{2}[(\mathbf{K}_{ss} + \mathbf{I}_{s^2})(\mathbf{I}_s \otimes \mathbf{Y}'\mathbf{L}_k)dvec(\mathbf{Y})] \qquad \mathbf{L}_k \text{ is symmetric}$$

$$= \frac{1}{2}[2\mathbf{N}_s(\mathbf{I}_s \otimes \mathbf{Y}'\mathbf{L}_k)dvec(\mathbf{Y})]$$

$$= \mathbf{N}_s(\mathbf{I}_s \otimes \mathbf{Y}'\mathbf{L}_k)dvec(\mathbf{Y})$$

$\square$

### 3.1.4 $Dc_k$

Putting it all together

$$Dc_k(\mathbf{Y}) = (2|\lambda_k^*|)sgn(\lambda_k^*)(\mathbf{u}_k' \otimes \mathbf{u}_k')(\mathbf{\Pi}_k' \otimes \mathbf{\Pi}_k')\mathbf{N}_s(\mathbf{I}_s \otimes \mathbf{Y}'\mathbf{L}_k) = vec\left(\frac{\partial c_k}{\partial Y}\right)'$$

We can simplify this to get the claim:

$$\frac{\partial c_k}{\partial Y} = (2|\lambda_k^*|)\text{sgn}(\lambda_k^*)\mathbf{L}_k\mathbf{Y}\mathbf{\Pi}_k\mathbf{u}_k\mathbf{u}_k'\mathbf{\Pi}_k'$$

*Proof.*

$$Dc_k(\mathbf{Y}) = (2|\lambda_k^*|)sgn(\lambda_k^*)(\mathbf{u}_k' \otimes \mathbf{u}_k')(\mathbf{\Pi}_k' \otimes \mathbf{\Pi}_k')\mathbf{N}_s(\mathbf{I}_s \otimes \mathbf{Y}'\mathbf{L}_k)$$

$$= (2|\lambda_k^*|)sgn(\lambda_k^*)(\mathbf{u}_k' \otimes \mathbf{u}_k')(\mathbf{\Pi}_k' \otimes \mathbf{\Pi}_k')\frac{1}{2}(\mathbf{K}_{ss} + \mathbf{I}_{s^2})(\mathbf{I}_s \otimes \mathbf{Y}'\mathbf{L}_k)$$

$$= (2|\lambda_k^*|)sgn(\lambda_k^*)\frac{1}{2}(\mathbf{u}_k'\mathbf{\Pi}_k' \otimes \mathbf{u}_k'\mathbf{\Pi}_k')(\mathbf{K}_{ss} + \mathbf{I}_{s^2})(\mathbf{I}_s \otimes \mathbf{Y}'\mathbf{L}_k)$$

$$= (2|\lambda_k^*|)sgn(\lambda_k^*)\frac{1}{2}\left[(\mathbf{u}_k'\mathbf{\Pi}_k' \otimes \mathbf{u}_k'\mathbf{\Pi}_k')\mathbf{K}_{ss}(\mathbf{I}_s \otimes \mathbf{Y}'\mathbf{L}_k) + (\mathbf{u}_k'\mathbf{\Pi}_k' \otimes \mathbf{u}_k'\mathbf{\Pi}_k')(\mathbf{I}_s \otimes \mathbf{Y}'\mathbf{L}_k)\right]$$

$$= (2|\lambda_k^*|)sgn(\lambda_k^*)\frac{1}{2}\left[(\mathbf{u}_k'\mathbf{\Pi}_k' \otimes \mathbf{u}_k'\mathbf{\Pi}_k')(\mathbf{Y}'\mathbf{L}_k \otimes \mathbf{I}_s)\mathbf{K}_{ns} + (\mathbf{u}_k'\mathbf{\Pi}_k' \otimes \mathbf{u}_k'\mathbf{\Pi}_k')(\mathbf{I}_s \otimes \mathbf{Y}'\mathbf{L}_k)\right]$$

$$= (2|\lambda_k^*|)sgn(\lambda_k^*)\frac{1}{2}\left[(\mathbf{u}_k'\mathbf{\Pi}_k'\mathbf{Y}'\mathbf{L}_k \otimes \mathbf{u}_k'\mathbf{\Pi}_k')\mathbf{K}_{ns} + (\mathbf{u}_k'\mathbf{\Pi}_k' \otimes \mathbf{u}_k'\mathbf{\Pi}_k'\mathbf{Y}'\mathbf{L}_k)\right]$$

$$= (2|\lambda_k^*|)sgn(\lambda_k^*)\frac{1}{2}\left[\mathbf{K}_{11}(\mathbf{u}_k'\mathbf{\Pi}_k' \otimes \mathbf{u}_k'\mathbf{\Pi}_k'\mathbf{Y}'\mathbf{L}_k) + (\mathbf{u}_k'\mathbf{\Pi}_k' \otimes \mathbf{u}_k'\mathbf{\Pi}_k'\mathbf{Y}'\mathbf{L}_k)\right]$$

$$= (2|\lambda_k^*|)sgn(\lambda_k^*)(\mathbf{u}_k'\mathbf{\Pi}_k' \otimes \mathbf{u}_k'\mathbf{\Pi}_k'\mathbf{Y}'\mathbf{L}_k) \qquad\qquad \mathbf{K}_{11} = 1$$

$$= (2|\lambda_k^*|)sgn(\lambda_k^*)(\mathbf{\Pi}_k\mathbf{u}_k \otimes \mathbf{L}_k\mathbf{Y}\mathbf{\Pi}_k\mathbf{u}_k)'$$

Then note that:

$$vec((2|\lambda_k^*|)sgn(\lambda_k^*)\mathbf{L}_k\mathbf{Y}\mathbf{\Pi}_k\mathbf{u}_k\mathbf{u}_k'\mathbf{\Pi}_k') = (2|\lambda_k^*|)sgn(\lambda_k^*)vec([\mathbf{L}_k\mathbf{Y}\mathbf{\Pi}_k\mathbf{u}_k][1][\mathbf{u}_k'\mathbf{\Pi}_k'])$$

$$= (2|\lambda_k^*|)sgn(\lambda_k^*)(\mathbf{\Pi}_k\mathbf{u}_k \otimes \mathbf{L}_k\mathbf{Y}\mathbf{\Pi}_k\mathbf{u}_k)vec(1)$$

$$= (2|\lambda_k^*|)sgn(\lambda_k^*)(\mathbf{\Pi}_k\mathbf{u}_k \otimes \mathbf{L}_k\mathbf{Y}\mathbf{\Pi}_k\mathbf{u}_k)$$

$$= (Dc_k(\mathbf{Y}))'$$

So that

$$\frac{\partial c_k}{\partial Y} = (2|\lambda_k^*|)\text{sgn}(\lambda_k^*)\mathbf{L}_k\mathbf{Y}\mathbf{\Pi}_k\mathbf{u}_k\mathbf{u}_k'\mathbf{\Pi}_k'$$

The proposition then follows by removing the absolute value and multiplication by the sign. □

Figure 1: The average number of neighbors $m(r)$ vs the neighborhood radius $r$, on a log-log scale, for the SDSS spectra data, computed on the whole sample of 675,000 galaxies. The blue regression line, is fitted to the graph points in the shown $r$ range, and has slope 2.87. The absence of a linear region on this graph suggests that the data dimension varies with the scale. The analysis and visualization in this paper corresponds to the largest meaningful scale.

## Footnotes

[1]see Matrix Differential Calculus With Applications in Statistics And Economics by Magnus & Neudecker ch. 9 §12 for proof