[Reviews · NeurIPS 2016]

Reviewer 1

Summary

The authors propose a practical algorithm for near-isometric embedding of manifold data. By leveraging from previous theoretical work, they develop an optimization problem that minimizes the difference between the Riemannian metric estimate of the embedded data and that of the given ambient data. Qualitative and quantitative experiments show that the proposed method achieves better performance than previous non-linear dimensionality reduction algorithms.

Qualitative Assessment

I like the overall approach of estimating the local Riemannian metric for achieving near isometric embedding of manifold data. Since the optimization criterion is non-convex, it would be nice to include a detailed discussion on - how initial embedding effects the final result - since the solution is iterative (gradient descent), quantitative analysis of how much slower is this approach compared to the standard eigendecomposition approaches. Minor comments: - I would recommend the authors to discuss the ‘NuMax’ algorithm in related work (citation: “NuMax: a convex approach for learning near-isometric linear embeddings” by Hedge et al.) - I dont quite understand what the authors meant by “relaxation” in the title. Consider changing it. - There are minor typos in lines 77, 82, and a possible square missing in the integral in Eq. 4.

Confidence in this Review

3-Expert (read the paper in detail, know the area, quite certain of my opinion)


Reviewer 2

Summary

The paper propose a manifold learning technique that seeks a low-dimensional embedded of data, such that it lies on a submanifold of the embedding space. This is achieved by gradient-based optimization of a straight-forward energy measuring the deviation of the intrinsic Riemannian metric from the identity. This is novel, but it is unclear why it is useful.

Qualitative Assessment

The key novelty of the paper is the idea that the intrinsic manifold dimension can (should?) be lower than the embedding dimension. This is novel, but the authors fail to motivate why it is useful. The only motivation seems to be that this approach can give less distortion. Some comments on this: 1) clearly lower distortions can be achieved when embedding a d-dimensional manifold into R^s (s > d) rather than embedding it into R^d (this is overly trivial); 2) it is not clear why we shouldn't just use Isomap (and its various cousins) to embed into R^s (this is what people do in practice); 3) it is unclear why this distinction is useful. I understand why it is helpful from a mathematical point of view, but which real problems can be solved that could not be solved before?

Confidence in this Review

2-Confident (read it all; understood it all reasonably well)


Reviewer 3

Summary

The paper introduces an algorithm to improve the low-dimensional embedding of data sampled from a manifold. The authors proposed distortion Loss functions, which are defined by the push-forward Riemannian metric associated with the given embedding coordinates. This loss directly measures the deviation from isometry. The initial embedding coordinates are then iteratively updated to reduce the loss. The loss function is shown to be non-convex. The new method--Riemannian Relaxation--was shown to have lower distortion than Laplacian Eigenmaps, MUV, Isomap, and HLLE on synthetic datasets.

Qualitative Assessment

The authors might need to explain more clearly on how the loss functions measure the deviation from isometry. If the loss is zero, does the embedding achieve isometry? Can you provide a theorem and proof on it? How to quantify the deviation from isometry if the loss is not zero? What is the quantitative measure for claiming that an embedding is nearly isometric? In addition, the loss function is non-convex, it might be quite difficult to achieve global optimal. In addition, the algorithm might be sensitive to the initialization (the initial coordinates Y^0). Is it crucial to initialize with the embedding coordinates from Laplacian Eigenmaps, Isomap, MVU, HLLE, or Diffusion maps? It might be useful to include the running times for different embedding methods. Riemannian Relaxation is a computationally more expensive method for doing embedding and the results shown in Figure 2 are not too different. The authors might need to provide more justification why it is useful in practice, for example, improving classification or regression results. The paper is well written overall. The organization is clear. There are a few typos in the paper, for example, Eq (7): what is $\mathcal{L}_k$?

Confidence in this Review

2-Confident (read it all; understood it all reasonably well)


Reviewer 4

Summary

This paper presents a new type of data embedding that preserves the geometry of the data as defined by a Reimannian metric (inner product on the tensor space of every point). Then they provide a method for gradient descent to solve this problem as the objective is not convex.

Qualitative Assessment

This paper claims that contrary to the plethora of embeddings it utilizes [3, 4,12,19,20] that the Reimannian metric is best for geometry preservation without motivating this thought or providing the intuition for this claim. The authors claim that unlike previous methods they do not provide a one-shot mapping, but they ignore the existence of methods like tsne which do employ gradient descent in deriving the embedding. Finally they do not test this on sufficiently many datasets of significant size show how their embeddings are better than previously existing datasets or are practical in terms of runtime. Figure 1: Their initialization is already very similar to the true shape; how does the algorithm perform without so much prior information? Also, the plot of MSE/loss vs sigma is poorly drawn. I'm not sure if they're constrained by runtime, but to connect the dots between their data points I would expect to see A) intervals for MSE & loss drawn from several randomized initializations and B) more points tested for sigma Figure 2: Poorly formatted; it's not immediately obvious that they have two data sets with the same embeddings. Also, top-left/top-right etc is difficult to follow; there should be sections of the figure labeled A-G with corresponding sections in the legend. Again, the line plots are not publication quality. There need to be more points sampled on the X axis or replicates of each condition to provide an interval; I assume the trends going up then down then up are an artifact of individual runs? Figure 3: Their analysis here is only half-complete. They present a real dataset, show superficial improvements in its analysis as a result of their method, but provide no physics-based analysis that would suggest their method actually improves the ability of scientists to analyze this kind of data.

Confidence in this Review

1-Less confident (might not have understood significant parts)


Reviewer 5

Summary

This paper proposes a new method of manifold learning which directly minimizes the deviation from isometry. The method is iteratively optimized by the projected gradient descent.

Qualitative Assessment

There are some comments as below: In algorithm 1, how to set d and s in practice? Is there a systematic way to select s once d is given? In line 25, please define the function poly. line 27: An embedding ... : please check the sentence line 71: W_{ij} --> W_{kl} In (1), D = W1 --> D = diag(W1), likewise, tilde(D) = diag(tilde(W) 1). line 77: give -- > gives In swiss roll and curved half sphere example (Fig. 2), the qualitative performance of HLLE and Isomap seems better. However, the numerical performance using (10) is favorable for the proposed RR. I want to see how different each of the two loss terms in (10) for different algorithms. The computational complexities of each algorithm are better be compared.

Confidence in this Review

2-Confident (read it all; understood it all reasonably well)


Reviewer 6

Summary

The authors propose an algorithm to embed a set of points in a low dimensional space. The algorithm creates an initial embedding and gradually improves it by minimizing a loss function. The proposed algorithm was experimentally evaluated on some tasks and it showed promising results.

Qualitative Assessment

The paper is very well-written and there are no obvious flaws. The authors clearly present the research questions they seek to answer as well as their contributions. The advance seems to be incremental. Specifically, the proposed algorithm is based on an existing estimator of the Laplace-Beltrami operator and an existing estimator of the push-forward metric. The results obtained by the proposed approach with regards to the state-of-the-art embedding algorithms are clearly presented. The authors have not made clear how to set the embedding dimension and intrinsic dimension parameters.

Confidence in this Review

1-Less confident (might not have understood significant parts)